# Impact of a New Preoperative Stratification Based on Cardiac Structural Compromise in Patients with Severe Aortic Stenosis Undergoing Valve Replacement Surgery [note 1]

**DOI:** 10.3390/diagnostics14192250

**Published:** 2024-10-09

**Authors:** Cristhian Espinoza Romero, Vitor Emer Egypto Rosa, Sérgio Octavio Kormann, Bryan Nicolalde, Antonio Sérgio de Santis Andrade Lopes, Guilherme Sobreira Spina, João Ricardo Cordeiro Fernandes, Flavio Tarasoutchi, Roney Orismar Sampaio

**Affiliations:** 1Instituto do Coração (InCor), Hospital das Clínicas HCFMUSP, Faculdade de Medicina, Universidade de São Paulo, São Paulo 05403-904, Brazil; vitoremer@yahoo.com.br (V.E.E.R.); crisviceromero1510@gmail.com (S.O.K.); antonio.santis@hc.fm.usp.br (A.S.d.S.A.L.); guilherme.spina@hc.fm.usp.br (G.S.S.); joao.fernandes@hc.fm.usp.br (J.R.C.F.); flavio.tarasoutchi@hc.fm.usp.br (F.T.); 2Norwalk Hospital-Yale University Program, Norwalk, CT 06856, USA; bryan.nicolalde@nuvancehealth.org

**Keywords:** aortic valve stenosis, aortic valve disease, postoperative complications

## Abstract

**Introduction and objectives:** Aortic valve replacement surgery (SAVR) remains a recommended indication, though its pre-surgical stratification is an ongoing challenge. Despite the widespread use of scores like the STS and EuroSCORE II, they have a number of limitations, while often neglecting structural parameters like left ventricular hypertrophy or left atrium volume. This study aimed to evaluate whether a new adaptation of the Généreux classification in the preoperative risk stratification of severe aortic stenosis (AS) is associated with the primary outcome, and to compare it with the original classification versus the traditional scores in short- and long-term follow-up. **Methods:** We conducted a retrospective, single-center study involving patients with confirmed severe AS who underwent SAVR. The new stratification categorized patients into three stages. Cox regression analyses were conducted to identify factors associated with mortality, with survival analysis performed using Kaplan–Meier curves. A *p*-value < 0.05 was considered statistically significant. **Results:** A total of 508 patients were included. Stage 3 patients had a lower median age (67 years). The median EuroSCORE II and STS scores were 2.75 and 2.62%, respectively (*p* ≤ 0.001). Over a median follow-up of 81 months, 56 deaths occurred (11%). Kaplan–Meier curve analysis revealed significant differences in all-cause mortality among the three groups (HR 4.073, log-rank *p* ≤ 0.001). Multivariable analysis identified the three preoperative stages (HR 3.22, [95% CI 1.44–7.20], *p* = 0.004) and mean transaortic gradient (HR 0.96, [95% CI 0.92–0.99], *p* = 0.021) as independent variables of mortality. The original Généreux scale AUC was higher (AUC: 0.760, 95% CI: 0.692–0.829) compared to the modified Généreux scale (AUC: 0.758, 95% CI: 0.687–0.829). However, no statistical differences were found between the different scales. **Conclusions:** Preoperative three-stage classification and low transaortic gradient are factors associated with increased all-cause mortality in patients undergoing SAVR. The proposed staging system performed better in the mortality analysis than EuroSCORE II and STS and was similar to the original classification.

## 1. Introduction

Aortic stenosis (AS) has become increasingly common, mainly due to the increase in life expectancy. In developed countries, the most common etiology of AS is degeneration, which mostly affects the elderly. However, in low- and middle-income countries, rheumatic heart disease remains a common cause of this condition [1,2,3]. Although Transcatheter Aortic Valve Replacement (TAVR) has shown similar or even better survival rates compared to Surgical Aortic Valve Replacement (SAVR), certain anatomical factors limit its use. Factors like a narrow valve annulus, extensive calcification, poor vascular access, or mismatched vascular diameters must be considered when deciding if TAVR is suitable [4]. Consequently, SAVR is often needed, presenting significant challenges in pre-surgical planning.

Risk assessment scores like the Society of Thoracic Surgeons (STS) score and the EuroSCORE II are widely used for cardiovascular surgeries. However, these scores have limitations for AS patients, as they often exclude structural factors like left ventricular (LV) hypertrophy, diastolic dysfunction (DD), left atrial (LA) volume, or secondary mitral or tricuspid regurgitation [5,6,7]. Another issue with existing scores (such as STS-PROM, EuroSCORE II, and TAVR-RS) is that they tend to overestimate mortality by 2–3 times in cases of minimally invasive SAVR [8]. Unlike AS, other valve disorders, like mitral regurgitation, commonly incorporate echocardiographic data in preoperative evaluations [1,2]. To address these gaps, Généreux et al. [9] proposed a classification system based on the degree of cardiac involvement, which was divided into five groups. However, the included patients had distinct epidemiological profiles, and the inclusion of both TAVR and SAVR procedures made the system less practical for clinical use. Despite its detailed anatomical stratification, we hypothesize that a simplified system could provide equally valuable information for predicting mortality outcomes. Additionally, most of the studies validating various risk scales have been conducted outside of Latin America.

This study aims to achieve three main objectives. The first is to evaluate and compare the performance of the STS, EuroSCORE II and Généreux scores, both original and modified, in the analysis of mortality in patients with SAVR. The second is to assess the epidemiology and outcomes of SAVR at one of Latin America’s largest centers, located in Brazil. The final objective is to propose a simplified version of the Généreux scale and compare its analysis of short- and long-term mortality with that of other scoring systems.

## 2. Materials and Methods

Population: A retrospective, single-center analysis involving 508 patients diagnosed with severe AS who underwent SAVR between January 2016 and December 2022 was performed (Figure 1). Considering an annual incidence of SAVR of 57,627, a sample of 382 individuals or more was necessary to acquire proper power and a confidence interval [10]. The definition of anatomically severe AS was established by the presence of an aortic valve area ≤ 1.0 cm^2^ and/or indexed ≤ 0.6 cm^2^/m^2^, along with a mean transaortic gradient ≥ 40 mmHg or maximum aortic jet velocity ≥4.0 m/s. In cases of AVA ≤ 1.0 cm^2^ and a mean LV/aorta gradient < 40 mmHg, a stress echocardiogram with dobutamine was performed. In those with ejection fraction (EF) <50% and in the absence of contractile reserve, a cardiac tomography with aortic valve calcium score was performed. In those with preserved EF, a calcium score was also performed. The exclusion criteria were patients with severe aortic regurgitation or other severe primary valvular diseases, primary cardiomyopathies, endocarditis, and severe coronary artery disease (defined as left main coronary artery lesion ≥ 50% or lesions in the three coronary vessels ≥ 70% with lesion in the proximal left anterior descending artery). The study protocol was reviewed and approved by the local institutional ethics committee. Informed consent was waived due to the retrospective nature of the study.

Data Collection: Clinical, laboratory, and echocardiographic data were collected and analyzed, along with variables related to the surgery, hospitalization, readmission, and mortality. The proposal was to reclassify the 5 original groups in a more simplified form, which is more applicable in daily practice. Each patient was classified into one of three groups based on the degree of progressive structural cardiac involvement, adapted from the classification by Généreux [9]. 

-Group 1: no or minimal cardiac damage (characterized by left ventricular [LV] impact with LVH [LV mass index > 95 g/m^2^ for women and >115 g/m^2^ for men]), DD (E/e′ > 14), or systolic dysfunction (LV ejection fraction [LVEF] < 50%);-Group 2: secondary/functional mitral valve dysfunction or LA dilation (defined by increased indexed volume [>34 mL/m^2^]), presence of atrial fibrillation (AF), or moderate to severe mitral regurgitation;-Group 3: pulmonary hypertension (PH), defined as systolic pulmonary artery pressure (SPAP) ≥ 60 mmHg, moderate to severe tricuspid regurgitation, or involvement of the right ventricle (RV) with moderate to severe dysfunction (tricuspid annular plane systolic excursion [TAPSE] <11 mm, Fractional area change [FAC] <25%).

The original classification into 5 groups is shown in Figure 2. 

Echocardiography data: Echocardiographic variables were assessed using methods recommended by current guidelines [11,12,13].

Outcomes: The primary endpoint was all-cause mortality, and the secondary endpoint included composite of cardiovascular mortality and cardiovascular readmissions. To calculate the outcomes, the median follow-up of the entire cohort was considered for the long term and at 30 days for the short term.

Statistical Analysis: Continuous variables were presented as a median (interquartile range) and categorical variables were presented as *n* (%). The Mann–Whitney test was used for continuous variables and Fisher’s exact test or Chi-square test was used for categorical variables. Post hoc analyses were performed using Tukey’s test. Kaplan–Meier curves and the log-rank test were used to assess all-cause mortality and the composite outcome of death and hospital readmission. Univariable and multivariable Cox regression analyses were employed to identify factors associated with the primary outcome. For the multivariable analysis, variables with a *p* < 0.10 in the univariable analysis were included. A significance level of *p* < 0.05 was considered statistically significant. The performance analysis of the modified classification was carried out and compared with the traditional scores and original classification using ROC curve analysis and area under the curve. An attempt was made to normalize the variables using a logarithmic transformation, but they continued to present an abnormal distribution, so nonparametric statistics were performed to establish the difference between groups. The criteria used to manage missing data were to apply the exclusion criteria described in the population section and to include only those patients who presented with at least 70% of the prespecified study variables to reduce possible confounding factors. All analyses were conducted using SPSS statistical software, version 21.

## 3. Results

Clinical and Laboratory Data: The clinical and laboratory characteristics are shown in Table 1. The median ages in stages 1, 2, and 3 were 62 (55–70), 67 (61–72), and 67 (60–73) years, respectively (<0.001). The EuroSCORE II for groups 1, 2, and 3 was 0.99 (0.88–1.30), vs. 1.43 (1.03–1.87) vs. 2.75 (1.63–3.81) %, respectively. In the post hoc analysis, a difference was found among the three groups (*p* < 0.001) in the same way as for the STS score which was 1.54 (1.44–1.64) vs. 2.00 (1.85–2.16) vs. 2.62 (2.25–2.99) % in the three stages, respectively, with a difference among the three groups (*p* < 0.001). Regarding brain natriuretic peptide (BNP), there were values of 79 (35–226) vs. 155 (81–331) vs. 920 (446–2740) pg/mL, respectively; and, in the post hoc analysis, the difference was among three stages (*p* < 0.001), as can be seen in Table 1.

Echocardiographic data: The echocardiographic characteristics are shown in Table 1. As expected, there is a significant difference among stages in relation to LV dysfunction (4.3% vs. 14.1% vs. 61.2%, *p* < 0.001), respectively. There was no significant difference among the groups in relation to the anatomical severity of AS, assessed through the aortic valve area [0.75 (0.60–0.85) vs. 0.7 (0.6–0.8) vs. 0.7 (0.6–0.85) cm^2^, respectively; *p* = 0.202]. However, the mean transaortic gradient [52 (43–63) vs. 53 (45–66) vs. 50 (40–60) mmHg, respectively; *p* = 0.023], there were differences between stages 2 and 3 in the post hoc analysis (*p* = 0.018). 

Postoperative data: The postoperative characteristics are shown in Table 1. Regarding all-cause mortality, there were also differences among the three stages (3.4% vs. 10.2% vs. 40.3%, respectively; *p* = 0.001), and this was also after subgroup analysis (*p* < 0.001). Concomitant surgeries are worth mentioning because they can influence postoperative outcomes; therefore, only patients undergoing myocardial revascularization surgery without a primary indication for it were included. Consequently, only 69 (13.6%) patients underwent revascularization. 

In the long-term analysis of the Kaplan–Meier curve, at a median follow-up of 81 (79–84) months, there were differences among the three groups regarding the primary outcome, as shown in Figure 2 [HR 4.073 (2.776–5.976), log-rank *p* ≤ 0.001]. In the other Kaplan–Meier curve at a median follow-up of 64 (61–68) months. Regarding composite outcomes by mortality and cardiovascular readmissions, there was a difference among the three groups, as shown in Figure 2 [HR 3.217 (2.552–4.056), log-rank *p* ≤ 0.001] (Appendix A). In the univariable analysis model excluding the variables contained in the preoperative stages, the variables associated with all-cause mortality were BNP, age, NYHA class, creatinine clearance, mean transaortic gradient, and proposed stage classification (Table 2). However, after multivariable analysis in a model that excluded the variables contained in the preoperative stages, the proposed stage classification (HR 3.22, [95% CI 1.44–7.20], *p* = 0.004) and the mean transaortic gradient (HR 0.96, [95% CI 0.92–0.99], *p* = 0.021) were the only independent variables of mortality (Table 2). Furthermore, in the multivariable analysis, multiple models were utilized, and even after adjusting for all variables including EuroSCORE II and STS, the same variables were obtained (Appendix A). The univariable and multivariable analyses were also performed based on the original Généreux classification, which was also a factor associated with mortality at the same follow-up time (Appendix A).

It is also important to mention that in the univariable analysis, parameters such as the presence of preoperative AF and SPAP as a continuous variable were associated with mortality (HR 4.467 [95% CI 2.438–8.186], *p* ≤ 0.001; HR 1.056, [95% CI 1.033–1.079], *p* ≤ 0.001, respectively), but were not included in the analysis because they were already included in the new classification.

Most traditional scores, including EuroSCORE II and STS, focus on the analysis of the short-term postoperative period, and therefore we carried out univariable and multivariable analyses that included these traditional calculators and the original and proposed classifications in a 30-day follow-up. The original and modified scales were the only factors associated with mortality after the multivariable analysis [HR 2.632; 95% CI (1.211–5.717); *p* = 0.015; HR 5019; 95% CI (1.596–15.784); *p* = 0.006] (Table 3).

Area under curve (AUC) analysis was performed to compare the different scales with regard to mortality outcome. The AUC at 30 days as well as during the entire follow-up period is represented in Figure 3. For the 30-day follow-up, the AUC of the original Généreux scale showed the best model (AUC: 0.815, 95% CI: 0.738–0.892) in comparison with the modified scale (AUC: 0.812, 95% CI: 0.732–0.892), EuroSCORE II (AUC: 0.722, 95% CI: 0.614–0.830), and STS (AUC: 0.695, 95% CI: 0.583–0.807), although the differences were not statistically significant as the confidence intervals overlapped among all scales (Appendix A). Regarding the entire follow-up period of 81 months (79–84), the original Généreux scale AUC was higher (AUC: 0.760, 95% CI: 0.692–0.829) compared to the modified Généreux scale (AUC: 0.758, 95% CI: 0.687–0.829), EuroSCORE II (AUC: 0.729, 95% CI: 0.663–0.794), and STS (AUC: 0.711, 95% CI: 0.649–0.773). However, no statistical differences were found between the different scales (Appendix A).

## 4. Discussion

The main findings of the present study are as follows: I. The proposed stage classification, which categorizes patients based on the presence or absence of extravalvular cardiac damage, was identified as an associated factor with mortality in a short- and long-term follow-up. This means that the greater the cardiac structural repercussions observed, the worse the prognosis after valve replacement surgery. II. A low transaortic gradient, potentially reflecting low-flow and low-gradient status, was also associated with a worse prognosis. III. The proposed staging classification demonstrated superior performance for mortality analysis than the EuroSCORE II and STS score, similar to the original classification.

In our study of the population with AS undergoing exclusively surgical intervention, it is important to highlight those patients classified in stage 3, who had a higher prevalence of the primary outcome, had a median age of 67 years. This is significantly lower compared to other studies, even those with low risk such as PARTNER 3 trial (with a mean age of 73 years) [3,5]. Therefore, there is a discrepancy between American and European guidelines, which recommend TAVR as the first-line treatment only in patients over 80 and 75 years of age, respectively, taking into account the surgical risk [1,2]. In our study, regarding the latter, the median STS score/EuroSCORE II was 2.62% and 2.75%, respectively, in stage 3, which differs from what is currently recommended in the guidelines, which is a surgical risk > 8% [1,2,3]. This opens the possibility of a hypothesis that our study generates, suggesting the need to consider TAVR in younger patients classified as stage 3, regardless of the scores mentioned above.

Furthermore, our study also raises the discussion about certain variables that were more prevalent in group 3 with a higher primary outcome, such as AF and SPAP. These variables were not included in the multivariable analysis because they were already included in the new classification, but they remain important associated factors for the stratification of our patients. Although parameters such as creatinine clearance and NYHA functional class were associated factors in the univariable analysis, they were not significant in the multivariable analysis. However, it is important to highlight that these are aspects that we consider in our daily clinical practice and must be considered when making clinical decisions.

AS is the most common valve disease, and its prevalence increases with age, reaching 11.8% in the population over 80 years old, mainly due to degenerative etiology [14]. However, in low-income countries, bicuspid and rheumatic etiologies remain prevalent [1,15]. Despite significant technological advances in surgical procedures and valve prostheses, complications after aortic valve replacement, including mortality, persist with reported 1-year mortality rates of 4.9% based on a European meta-analysis [16]. There are other trials with long-term follow-up, with a 5-year mortality rate of 16% for low surgical risk measured by EuroSCORE II, similar to data found in Brazil of 16.1% at the same follow-up time [17,18]. 

There are limited scores with adequate accuracy available to predict intra- and postoperative outcomes in valve replacement, and their applicability remains challenging due to population heterogeneity [17]. EuroSCORE II and STS are the most widely used and validated scores globally, playing a crucial role in cardiac surgical practice by assessing postoperative mortality and morbidity. However, EuroSCORE II tends to overestimate mortality in low-risk patients and underestimate it in high-risk patients [19]. The STS score is more complex, comprising over 40 clinical parameters [19]. Both scores have limitations as they were not specifically designed for low- and middle-income countries, and mainly include variables specific to the individual rather than cardiac involvement secondary to aortic valve disease [15,20,21]. Another limitation of existing risk scores (such as STS, EuroSCORE II, and TAVR-RS) is that they overestimate mortality by 2–3 times in the case of minimally invasive SAVR [8]. Furthermore, they primarily predict short-term events, and there is a lack of comprehensive studies with long-term follow-up [22,23]. To address this gap, there is an urgent need for new evaluation tools focused on the cardiac repercussions of this severe valve disease. Several studies have already assessed structural impairment through echocardiography, correlating it with a worse postoperative prognosis [9,10].

Généreux et al. proposed a five-stage classification system based on the presence or absence of extravalvular cardiac damage [9]. Although this classification demonstrated a good performance ability, it is complex due to its reliance on five stages and was not validated in the Latin American population, which may have a different epidemiological profile.

In our study, we aimed to simplify this classification and evaluate it in an exclusively surgical population. Notably, patients who underwent TAVR, accounting for approximately 65% of the original study’s cohort, were not included. Furthermore, we consolidated stages 0 and 1 into a single group (current group 1) and stages 3 and 4 into another group (current group 3) due to their low prevalence in both the current study (5.4% for stage 4) and the original study (8.7% for stage 4). Our adapted classification emerged as an independent associated factor of poor prognosis for postoperative mortality after valve replacement. The higher the stage (group 1 to 3), the more extensive the cardiac involvement, leading to a greater number of adverse events. Additionally, we found that the EuroSCORE II and STS score failed to predict long-term mortality and significantly underestimated mortality, particularly in groups 2 and 3 as per the proposed stage classification (all-cause mortality in stage 2 (10.2%) vs. EuroSCORE (1.43%) vs. STS score (2.0%), and the stage 3 (40.3%) vs. EuroSCORE (2.75%) vs STS (2.62%).

The early randomized trials comparing SAVR and TAVR were conducted in patients with high surgical risk, defined by an STS score >10% or a 30-day mortality risk >15%, as estimated by the Heart Team prior to surgery [5,24]. For those with intermediate risk, an STS score of 4–8% or a Heart Team estimate between 3 and 15% was applied [24,25]. Lastly, recent low-risk trials used a 30-day mortality risk <3%, or an STS score <4%, as estimated by the local Heart Team [26,27]. When comparing these data with our cohort, we observe that, according to traditional scoring systems, our population presents an intermediate to low risk considering their age, functional class, and scores. However, despite this intermediate age and the low scores, the mortality rate in group 3 is high, highlighting the role of anatomical factors in this population with different epidemiological characteristics than the other trials. Other recent adaptations and classifications have been applied to patients with AS, particularly those undergoing TAVR. One such adaptation included asymptomatic patients, where 14% of the population was classed as at least stage 3 or 4, further underscoring the utility of anatomical stratification, as staging was significantly associated with excess mortality in a multivariable analysis adjusted for SAVR as a time-dependent variable [28]. In another study, which evaluated 262 patients with severe AS, a gradual increase in mortality rates was observed over a one-year follow-up, with 18.6% mortality in stage 3 and 21.6% in stage 4 [29]. However, the STS score was >6% in stages 3 and 4, which was quite different from the scores found in our population. Therefore, our study provides important evidence, showing that, just like in the aforementioned populations, anatomical stratification has significant value in our cohort regardless of symptoms, risk scores, or clinical variables such as age.

Another noteworthy prognostic factor was the mean transaortic gradient, which was consistent with findings from other studies [30,31]. This prompted us to consider the prognostic implications of low-flow and low-gradient AS, a subgroup with a worse prognosis compared to those with high-gradient AS [32,33]. Recent studies have already demonstrated the cardiac structural impact that this entity causes in the LV, including more interstitial fibrosis than patients with high-gradient AS [34,35].

An important point to mention is that both the original and proposed classifications include the presence of PH with a systolic cut-off value of >60 mmHg among their variables. When tested as a continuous variable in the present study, PH was an associated factor in the univariable analysis. However, it was not evaluated in the multivariable analysis because this variable is already embedded within the scales. This raises the issue that some patients with PH who do not reach the >60 mmHg threshold may still experience worse postoperative outcomes, as demonstrated in some studies and within this specific population [36,37].

One of the strengths of this study is its potential clinical applicability. An important question that remains unanswered is whether TAVR is preferable to SAVR in patients with PH and/or RV dysfunction. This question cannot be addressed by the two previous studies because they only included patients undergoing TAVR [38,39]. However, recent analyses from the PARTNER IIA (Placement of AoRTic TraNscathetER Valves IIA) randomized trial reported that worsening RV function was four times more frequent following SAVR than TAVR, and this was associated with a two-fold increase in the risk of mortality [40]. This highlights another strength of our study, as this classification has not yet been applied to an exclusively surgical population. Considering that a significant percentage of patients with severe AS (20% in asymptomatic severe AS and up to 50% in symptomatic severe AS) have advanced cardiac damage, classed as at least stage 3, and are at markedly higher risk of mortality after intervention, the emergence of such right-sided abnormalities should prompt early consideration of intervention. Additionally, the presence of this anatomical impairment strengthens the argument in favor of selecting TAVR over SAVR in these cases. Another important strength of our study, as previously mentioned, is that our population differs by being younger, with lower biomarker levels, predominantly NYHA class II, and lower preoperative traditional scores. Regardless of these factors, as well as in other populations where a similar system was applied, anatomical stratification remains an important variable and should be considered in preoperative evaluations. Simplified into three stages, this stratification is an important factor for both short- and long-term mortality analysis, making it an invaluable tool for guiding treatment decisions. For instance, in group 3, where mortality was higher regardless of traditional scores or age, a percutaneous approach would likely be the most appropriate intervention. Another of the possible clinical applications of the present study is that in this group 3 population with high mortality for SAVR, interventions such as TAVR or mini SAVR would be applicable to this population.

However, there are several limitations inherent to this retrospective analysis. For instance, comorbidities like diabetes, arterial hypertension, and CAD were not excluded, which may introduce confounding factors. However, all these variables were included in the univariable and multivariable analyses to reduce potential confounding factors. In addition, the prevalence of CAD was low and patients with severe CAD were not included. Another potential limitation is that the right-sided abnormalities may not be directly caused by AS per se but may rather be related to other comorbidities, including pulmonary and coronary artery disease. Furthermore, while the study adjusts for some variables, not all potentially important biomarkers like troponin were included. Although our population is not very old, it is important to consider that along with aging there is calcification of the mitral annulus, and the current classification evaluates diastolic function through the E/e’ ratio, and that due to this calcification, it can be an erroneous measurement. Our study did not include factors of frailty, but it is worth emphasizing that our population is younger. It is important to mention that this article provides another example that there are multiple characteristics that make up the preoperative risk of a patient undergoing cardiac surgery; however, for future studies these preoperative stratification models can improve with some adjustments, as commented by Griffin and colleagues, aiming at greater surveillance during the postoperative period [41].

## 5. Conclusions

The main conclusions drawn from this study were that the new proposed stage classification and the low transaortic gradient are factors associated with all-cause mortality and the composite outcome by mortality and cardiovascular readmissions during short- and long-term follow-up in patients undergoing SAVR. The proposed staging performed better for mortality analysis than EuroSCORE II and STS and was similar to the original classification.

## Figures and Tables

**Figure 1 diagnostics-14-02250-f001:**
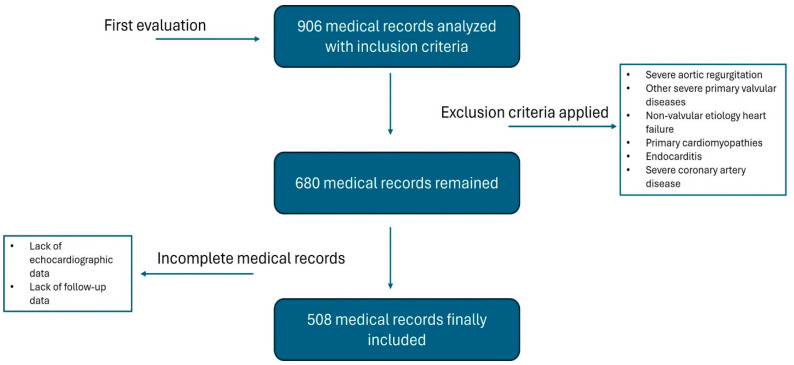
Flowchart of patients included.

**Figure 2 diagnostics-14-02250-f002:**
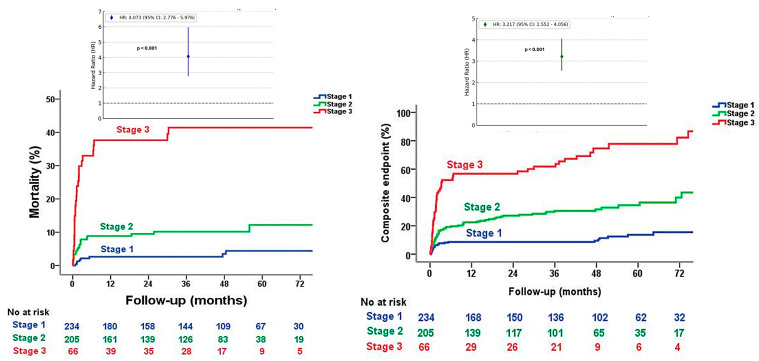
Kaplan–Meier curve of time to all-cause mortality outcome in the proposed stage classification and hazard ratio with confidence interval on the left. Kaplan–Meier curve of time to composite outcome by mortality and readmission in the proposed stage classification and hazard ratio with confidence interval on the right.

**Figure 3 diagnostics-14-02250-f003:**
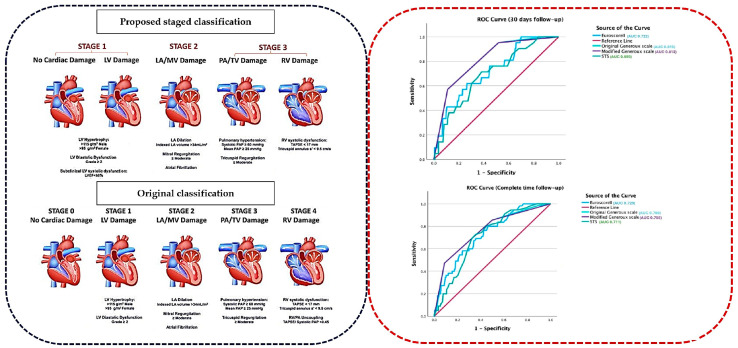
ROC curve analysis of the STS, EuroSCORE II and proposed and original scale classification for long-term (left) and short-term (right) follow-up time.

**Table 1 diagnostics-14-02250-t001:** Baseline clinical, laboratory, and echocardiographic data and postoperative findings.

Variable	Stage 1 (N = 235)	Stage 2 (N = 206)	Stage 3 (N = 67)	*p*
**Clinical data**				
Age, years	62 (55–70)	67 (61–72)	67 (60–73)	<0.001 †‡
Male sex	139 (59.1)	110 (53.4)	41 (61.2)	0.365
Diabetes Mellitus	66 (28.1)	77 (37.4)	24 (35.8)	0.100
Arterial hypertension	155 (66.0)	158 (76.7)	49 (73.1)	0.042 †
Dyslipidemia	151 (64.3)	138 (67.0)	40 (59.7)	0.542
Chronic kidney disease	21 (8.9)	20 (9.7)	14 (20.9)	0.017 ‡¥
Atrial fibrillation	1 (0.4)	22 (10.7)	17 (25.4)	<0.001 †‡¥
Coronary artery disease				0.346
One vessel	42 (17.9)	30 (14.6)	12 (17.9)	
Two vessels	15 (6.4)	24 (11.7)	9 (13.4)	
Rheumatic etiology	13 (5.5)	17 (8.3)	5 (7.5)	0.158
Bicuspid etiology	37 (15.7)	18 (8.7)	6 (9.0)	0.157
Diuretic use	144 (61.3)	161 (78.2)	58 (86.6)	<0.001 †‡
ACE-I use	61 (26.0)	53 (25.7)	33 (49.3)	<0.001 †‡
ARB use	77 (32.8)	78 (37.9)	13 (19.4)	0.020 ¥
B-blocker use	35 (14.9)	55 (26.7)	29 (43.3)	<0.001 †‡¥
Statin use	153 (65.1)	137 (66.5)	40 (59.7)	0.597
EuroSCOREII, %	0.99 (0.80–1.30)	1.43 (1.03–1.87)	2.75 (1.63–3.81)	<0.001 †‡¥
STS score, %	1.54 (1.44–1.64)	2.00 (1.85–2.16)	2.62 (2.25–2.99)	<0.001 †‡¥
Symptoms				
NYHA				<0.001
II	129 (54.9)	90 (43.7)	6 (9.0)	‡¥
III	79 (33.6)	95 (46.1)	48 (71.6)	†‡¥
IV	7 (3.0)	10 (4.9)	13 (19.4)	‡¥
**Laboratory data**				
Glomerular filtration rate, mL/min ^2,^*	72 (62–87)	65 (56–80)	58 (48–70)	<0.001 †‡¥
Hemoglobin, g/dL ^1,^*	13.3 (12.3–14.5)	13.0 (11.7–14.2)	13.0 (11.8–14.2)	0.033 †
BNP, pg/mL ^96,^*	79 (35–226)	155 (81–331)	920 (446–2740)	<0.001 †‡¥
**Echocardiographic data**				
Aortic regurgitation (mild or moderate) ^2,^*	110 (47.0)	136 (66.3)	58 (86.6)	<0.001 †‡
Left Ventricular Hypertrophy ^2,^*	165 (70.2)	191 (92.7)	60 (89.6)	<0.001 †‡
Diastolic dysfunction	146 (62.1)	182 (88.3)	43 (64.2)	<0.001 †¥
Ejection fraction < 50%	10 (4.3)	29 (14.1)	41 (61.2)	<0.001 †‡¥
Enlargement left atrium	10 (4.3)	200 (97.1)	67 (100.0)	<0.001 †‡
Mitral regurgitation (moderate/severe)	2 (0.9)	45 (21.8)	32 (47.8)	<0.001 †‡¥
Pulmonary hypertension ^107,^*	0 (0.0)	0 (0.0)	34 (50.7)	<0.001 ‡¥
Tricuspid regurgitation moderate/severe	1 (0.4)	0 (0.0)	43 (64.2)	<0.001 ‡¥
**RV** dysfunction (moderate/severe)	0 (0.0)	0 (0.0)	12 (17.9)	<0.001 ‡¥
Mass ^2,^*, g/m^2^	120 (100–140)	130 (114–154)	142 (124–179)	<0.001 †‡¥
Mean gradient aortic, mmhg	52 (43–63)	53 (45–66)	50 (40–60)	0.023 ¥
Aortic valvular area ^1,^*, cm^2^	0.75 (0.60–0.85)	0.7 (0.6–0.8)	0.7 (0.60–0.85)	0.202
Pulmonary systolic pressure, mmhg ^107,^*	30 (26–36)	35 (30–42)	60 (50–65)	<0.001 †‡¥
Ejection fraction, %	63 (60–66)	62 (57–66)	45 (30–60)	<0.001 ‡¥
**Postoperative findings**				
Postoperative AF	44 (18.7)	52 (25.2)	28 (41.8)	0.001 ‡¥
Long-term all-cause mortality	8 (3.4)	21 (10.2)	27 (40.3)	<0.001 †‡¥
Long-term all-cause mortality and cardiovascular readmissions	26 (11.1)	65 (31.6)	51 (77.3)	<0.001 †‡¥
30-day mortality	1 (0.4)	10 (4.9)	15 (22.7)	<0.001 †‡¥

Values as median (interquartile range) or n (%). NYHA: New York Heart Association. AF: Atrial fibrillation. † Significant difference (*p* < 0.05) between groups 1 and 2. ‡ Significant difference (*p* < 0.05) between groups 1 and 3. ¥ Significant difference (*p* < 0.05) between groups 2 and 3. * Number of missing data for each variable; if it does not have the symbol * then no values are missing.

**Table 2 diagnostics-14-02250-t002:** Univariable and multivariable analysis of all-cause mortality.

Variable	Univariable Analysis		Multivariable Analysis	
	HR (CI 95%)	*p*	HR (CI 95%)	*p*
Proposed stage classification	4.07 (2.77–5.97)	**<0.001**	3.22 (1.44–7.20)	**0.004**
Mean transaortic gradient, mmHg	0.97 (0.95–0.98)	**0.003**	0.96 (0.92–0.99)	**0.021**
Glomerular filtration rate, mL/min	0.96 (0.95–0.97)	**<0.001**	0.99 (0.96–1.01)	**0.431**
BNP, pg/mL	1.000 (1.000–1.001)	**0.001**	1.000 (1.000–1.001)	**0.172**
Age, years	1.04 (1.01–1.07)	**0.003**	1.03 (0.98–1.09)	**0.168**
NYHA stages	2.72 (1.83–4.05)	**<0.001**	0.86 (0.40–1.82)	**0.701**

**Table 3 diagnostics-14-02250-t003:** Univariable and multivariable analysis of 30-day mortality analysis of original and proposed classification.

Univariable Analysis					Multivariable Analysis				
					Original Classification				
Variables	HR	CI, 95%		*p*	Variables	HR	CI, 95%		*p*
**Proposed staged classification**	5.655	3.034	10.539	<0.001	BNP	1.000	1.000	1.000	0.629
					EuroSCORE II	1.004	0.677	1.490	0.983
**Original classification**	3.247	2.165	4.870	<0.001	STS score	0.953	0.602	1.509	0.839
					Creatinine clearance	0.985	0.949	1.022	0.431
**BNP**	1.000	1.000	1.001	0.004	Original classification	2.632	1.211	5.717	0.015
					NYHA stages	0.551	0.178	1.705	0.301
**EuroSCORE II**	1.278	1.127	1.450	<0.001	Proposed staged classification				
					BNP	1.000	1.000	1.001	0.533
**STS score**	1.274	1.006	1.613	0.045	EuroSCORE II	1.019	0.678	1.531	0.929
					STS score	0.901	0.558	1.456	0.671
**Creatinine clearance**	0.965	0.945	0.985	0.001	Creatinine clearance	0.986	0.950	1.024	0.470
					NYHA stages	0.474	0.150	1.493	0.202
**NYHA stages**	2.638	1.501	4.637	0.001	Proposed classification	5.019	1.596	15.784	0.006


## Data Availability

The original contributions presented in the study are included in the article, further inquiries can be directed to the corresponding authors.

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
