# Peer review of "Impact of a New Preoperative Stratification Based on Cardiac Structural Compromise in Patients with Severe Aortic Stenosis Undergoing Valve Replacement Surgery†"

_diagnostics, 2024, doi:10.3390/diagnostics14192250_

Round 1

Reviewer 1 Report

Comments and Suggestions for Authors

This study presents a novel, simplified classification system for preoperative stratification of severe aortic stenosis patients undergoing valve replacement surgery. It improves upon existing models by incorporating structural cardiac parameters that are often overlooked in traditional scoring systems. Although the proposed classification system shows promising results in predicting mortality, the study could further discuss its comparative strengths and weaknesses relative to other models and expand on its potential clinical utility. The manuscript demonstrates good potential but requires minor revisions to strengthen its impact.

Strengths

The manuscript provides a thorough analysis of preoperative stratification in patients with severe aortic stenosis undergoing valve replacement surgery. The introduction clearly outlines the limitations of traditional scoring systems like STS and Euroscore II in predicting outcomes, especially by excluding structural parameters like left ventricular hypertrophy and left atrium volume. The methods section is well-detailed, and the proposed simplified adaptation of the Généreux classification addresses these limitations effectively. The study’s innovation lies in its simplification of an established classification system to improve practical application in preoperative risk stratification. The new three-stage classification demonstrates comparable predictive ability to the original Généreux classification, but with more ease of use in clinical settings. This simplified stratification can help to refine mortality prediction and support clinical decision-making, especially in populations not previously well studied, such as those in Latin America. However, the study's novelty is somewhat diminished by the fact that it primarily tests a modified version of an existing system rather than proposing a wholly new methodology.

Weaknesses and Improvement

1.While the study builds on the Généreux classification, there could be more discussion regarding how the modified classification compares with other contemporary classification systems or newer risk models beyond Euroscore and STS.

2.The manuscript would benefit from a more detailed discussion on the clinical implementation of the modified classification system, including examples of how it could change treatment decisions in comparison to traditional scoring systems.

3.The Kaplan-Meier survival curves and ROC curves are useful, but additional visual aids (e.g., a flowchart comparing the traditional and modified classification processes) could enhance the reader's understanding of how the simplified classification works in practice.

4.The limitations section does not sufficiently address the retrospective nature of the study, which can introduce bias. This could be expanded to include more discussion on potential confounders and how they were managed, especially given the wide range of clinical presentations and comorbidities in the patient cohort.

Comments on the Quality of English Language

The quality of the English language in this manuscript is generally good, with clear and concise explanations of the research methodology and findings. However, there are a few areas where minor grammatical improvements and rewording for clarity may enhance readability.

Author Response

3. Point-by-point response to Comments and Suggestions for Authors

Comments 1: 1.While the study builds on the Généreux classification, there could be more discussion regarding how the modified classification compares with other contemporary classification systems or newer risk models beyond Euroscore and STS.

Response 1: We agree with this comment. I am very grateful for the valuable comment, a paragraph with further discussion was included between lines 289 and 328.

“The early randomized trials comparing SAVR and TAVR were conducted in patients with high surgical risk, defined by an STS score >10% or a 30-day mortality risk >15%, as estimated by the Heart Team prior to surgery.5,23 For those with intermediate risk, an STS score of 4%-8% or a Heart Team estimate between 3%-15% was applied.23,24 Lastly, recent low-risk trials used a 30-day mortality risk <3%, or an STS score <4%, as estimated by the local Heart Team.25,26 When comparing these data with our cohort, we observe that, according to traditional scoring systems, our population presents an intermediate to low risk, considering their age, functional class and scores. However, despite this intermediate age and low scores, mortality in group 3 is high, highlighting the role of anatomical factors in this population. Other recent adaptations and classifications have been applied to patients with AS, particularly those undergoing TAVR. One such adaptation included asymptomatic patients, where 14% of the population had at least stage 3 or 4, further underscoring the utility of anatomical stratification, as staging was significantly associated with excess mortality in a multivariate analysis adjusted for SAVR as a time-dependent variable.27 In another study, which evaluated 262 patients with severe AS, a gradual increase in mortality rates was observed over a one-year follow-up, with 18.6% of mortality in stage 3 and 21.6% in stage 4.28 However, the STS score was >6% in stages 3 and 4, quite different from our population. Therefore, our study provides important evidence, showing that, just like in the populations, anatomical stratification has significant value in our cohort, regardless of symptoms, risk scores, or clinical variables such as age.”

Comments 2: The manuscript would benefit from a more detailed discussion on the clinical implementation of the modified classification system, including examples of how it could change treatment decisions in comparison to traditional scoring systems.

Response 2: We agree with this comment. I am very grateful for the valuable comment, a paragraph with further discussion was included between lines 343 and 367.

One of the strengths of this study is its potential clinical applicability. One im-portant question that remains unanswered is whether TAVR is preferable to SAVR in patients with PH and/or RV dysfunction. This question cannot be addressed by the two previous studies because they only included patients undergoing TAVR.30,31 However, recent analyses from the PARTNER IIA (Placement of AoRTic TraNscathetER Valves IIA) randomized trial reported that worsening RV function was four times more frequent following SAVR than TAVR, and this was associated with a two-fold increase in the risk of mortality.32 This highlights another strength of our study, as this classification has not yet been applied to an exclusively surgical population. Considering that a significant percentage of patients with severe AS (20% in asymptomatic severe AS and up to 50% in symptomatic severe AS) have advanced cardiac damage, at least stage 3, and are at markedly higher risk of mortality after intervention, the emergence of such right-sided abnormalities should prompt early consideration of intervention. Additionally, the presence of this anatomical impairment strengthens the argument in favor of selecting TAVR over SAVR in these cases. Another important strength of our study as previously mentioned, our population differs by being younger, with lower biomarker levels, pre-dominantly NYHA class II, and lower preoperative traditional scores. Regardless of these factors, as well as in other populations where a similar system was applied, ana-tomical stratification remains an important predictor and should be considered in pre-operative evaluations. Simplified into three stages, this stratification is a strong predictor of both short- and long-term mortality, making it an invaluable tool for guiding treat-ment decisions. For instance, in group 3, where mortality was higher regardless of tra-ditional scores or age, a percutaneous approach would likely be the most appropriate intervention. Another of the possible clinical applications of the present study is that in this group 3 population with high mortality for SAVR, interventions such as TAVR or mini SAVR would be applicable to this population.

Comments 3: The Kaplan-Meier survival curves and ROC curves are useful, but additional visual aids (e.g., a flowchart comparing the traditional and modified classification processes) could enhance the reader's understanding of how the simplified classification works in practice.

Response 3: Agree. We have, accordingly, and the text was modified. The figure was added in the paper.

Comments 4: The limitations section does not sufficiently address the retrospective nature of the study, which can introduce bias. This could be expanded to include more discussion on potential confounders and how they were managed, especially given the wide range of clinical presentations and comorbidities in the patient cohort.

Response 4: Thank you very much for the comment. The limitations were expanded and placed in the text. See in lines between 369-376.

“Comorbidities like diabetes, arterial hypertension, and CAD were not excluded, which may introduce confounding factors. However, all these variables were included in the univariate and multivariate analysis to reduce potential confounding factors. In addition, the prevalence of CAD was low and patients with severe CAD were not included. Another potential limitation is that the right-sided abnormalities may not be directly caused by AS per se but may rather be related to other comorbidities including pulmonary and coronary artery disease. Furthermore, while the study adjusts for some variables, not all potentially important biomarkers like troponin were included”.

4. Response to Comments on the Quality of English Language

Point 1: The quality of the English language in this manuscript is generally good, with clear and concise explanations of the research methodology and findings. However, there are a few areas where minor grammatical improvements and rewording for clarity may enhance readability.

Response 4: Thanks for the feedback, English has improved overall.

Reviewer 2 Report

Comments and Suggestions for Authors

The manuscript titled "Impact of a New Preoperative Stratification Based on Cardiac Structural Compromise in Patients with Severe AS Undergoing AVR" by Romero et al., from Brazil, presents a study evaluating a modified version of the Généreux classification. The authors aimed to simplify the classification for practical use and assess its predictive ability for all-cause mortality in patients undergoing surgical aortic valve replacement (SAVR). The study compares the predictive accuracy of this new stratification with the original Généreux classification and traditional scoring systems (STS and Euroscore II) in both short- and long-term follow-up.

Major Comments:

  1. Introduction:

a.        The authors should have cited the study by Alnajar et al. (Innovations 2021, DOI: 10.1177/1556984520971775) in both the introduction and discussion, as it directly demonstrates that existing risk scores (e.g., STS-PROM, EuroSCORE II, TAVR-RS) overestimate mortality by 2-to-3-fold for minimally invasive AVR. This aligns with the current study’s focus on refining preoperative stratification for SAVR and would highlight similar findings in risk score inaccuracies, and in the discussion, it would reinforce the argument for recalibrating existing models for newer surgical techniques​.

  1. Statistics/Methods:

a.        The cohort has STS scores below 4% without a detailed rationale for this exclusion high risk patients. Excluding higher-risk populations can skew comparisons with registry-based risk scores, which reflect the entire spectrum of patient risks​. Including an explanation for this exclusion in the manuscript, and potentially analyzing the outcomes of this excluded group separately, would help clarify whether their mortality rates were significantly different or if they could have been included in the broader analysis.

b.       While the statistical methods used (e.g., Kaplan-Meier, Cox regression) are appropriate for the study, more transparency is needed regarding the failure to normalize certain variables. The manuscript should explain why normalization was not successful and if other methods of transformation could have been used but were not considered for a potential impact to the results.

c.        Although the authors performed multivariate analysis to adjust for potential confounders, diseases such as diabetes and coronary artery disease were not excluded from the cohort. This introduces a risk of confounding, especially considering their prevalence in the study population. Addressing how these comorbidities may have affected mortality predictions would enhance the robustness of the findings.

d.       The authors mention using SPSS statistical software for data analysis, but further clarification on how missing data was handled would be helpful. It is important to address missing data in retrospective analyses. Were there specific variables missing that might have impacted the accuracy of the risk calculations?

  1. Results:

a.        The results section should not be introduced within the methods section. In this manuscript, some descriptions of results (e.g., mortality outcomes and patient demographics) appear prematurely in the methods section, which should be focused solely on the study design, statistical analysis, and procedures. Moving these findings to the results section would improve the manuscript’s structure and ensure clarity and proper flow.

b.       The authors mentioned "prediction" in their methods, but the paper does not seem to include any formal internal validation or predictive modeling to support these claims. Proper prediction modeling typically requires data splitting (e.g., into training and testing sets) or cross-validation. Perhaps I have overlooked it, so please either direct me to the section where prediction modeling was used, or revise the terminology to "association" to reflect the analyses conducted in the study if no prediction modeling was performed.

c.        BNP levels showed significant differences across stages, but the multivariate analysis did not find BNP to be an independent predictor. A deeper exploration of why BNP’s predictive value diminished in the multivariate analysis would help clarify its role in the proposed classification system.

d.       Although the manuscript states that there were no statistical differences in the ROC curves between the scales, adding more detailed confidence interval comparisons might help clarify the magnitude of the differences. 95% CI will not only show the direction and strength of the demonstrated effect but will also provide a range in which the true value lies.

e.        “Complications” was one of the keywords of the manuscript but nor mentioned within the manuscript. In the results, I can only see postop Afib. Reporting major complications such as reentry for bleeding, major wound infections, or paravalvular leaks were not extensively discussed. While the current manuscript focuses heavily on mortality, including these complication rates would add depth to the outcomes, especially in comparison to more traditional SAVR and TAVR techniques, where such complications might be more prevalent.

  1. Discussion:

a.        It would strengthen the discussion to acknowledge that, while the current study focuses on SAVR, the broader surgical landscape includes comparisons to less invasive techniques like mini-AVR and TAVR. These comparisons are critical when discussing mortality and outcomes. For instance, in the mini-AVR population (Alnajar et al. Innovations 2021), it was pointed out that risk scores should ideally reflect differences between traditional and minimally invasive techniques. The authors could also explicitly discuss how their findings might apply to patients undergoing TAVR and if their proposed classification could be generalized to this population.

b.       The modified Généreux classification demonstrated similar predictive ability as the original scale, but with fewer stages. However, as this study was performed in a single center in Brazil, thus external validation in larger multicenter datasets, particularly considering that different patient populations may present with different epidemiological characteristics other populations is necessary. Please include a discussion about the generalizability of the model to different ethnic and geographic populations for added important value.

c.        The study is based on a single-center data, which can introduce bias due to the experience and skill level of individual surgeons. Please address mechanisms in your center for ensuring accurate and complete data capture, including oversight mechanisms and data auditing​. It would be useful particularly since this is a retrospective study and single-institution outcomes can sometimes reflect individual practice nuances.

Minor Comments:

  1. Figure Improvements: The flowchart (Figure 1) and Kaplan-Meier curves (Figure 2) are well-constructed, but more visual clarity can be provided by including exact p-values and shaded confidence limits within the figure. The confidence limits are better displayed as a colored shaded area or bars in the figure are preferred, but both could be added as a table in the figure.
  2. Limitations Section: Although limitations are discussed, it would benefit from a stronger focus on the potential selection bias due to the retrospective nature of the study. Furthermore, while the study adjusts for some variables, not all potentially important biomarkers (e.g., troponin) were included.
  3. Language and Grammar: Minor grammatical issues are present, such as in the introduction and conclusion. A thorough proofreading would enhance the manuscript’s readability.
  4. The authors mentioned using a multivariable cox. I just would like to kindly point out that multivariable ≠ multivariate. This an overwhelmingly common mistake that we all should be aware of in our research. For more clarification please read: Hidalgo and Goodman 2013"Multivariate or multivariable regression?" Am J Public Health doi: 10.2105/AJPH.2012.300897. PMC3518362.

Comments on the Quality of English Language

In addition to the above minor comments, here are some explicit suggestions aiming to improve the mansucript:

 ·  Grammar and Syntax(page 2) There are several instances of awkward phrasing and grammar mistakes throughout the manuscript. These should be corrected for clarity and flow. For example, reword sentences with overly complex structures or unclear references.

·  Tense Consistency(page 4) The manuscript inconsistently switches between past and present tense, especially when discussing study results and procedures. Ensure past tense is used for methods and results, and present tense for general findings or literature references.

·  Repetitive Phrasing(page 7) Some phrases and terms are repeated unnecessarily, particularly in the introduction and discussion sections. Eliminating redundancy would improve readability.

·  Word Choice(page 6) Certain terms, such as "prediction," "validate," and "significant," are used incorrectly or too loosely in several places. More precise word choice is needed, especially for scientific and statistical concepts.

·  Passive Voice Overuse(page 5) The manuscript often relies on passive voice, which can obscure the subject and make sentences less engaging. Switching to active voice where appropriate would improve clarity.

·  Article Use (a/the)(page 3) Incorrect use of definite and indefinite articles ("a," "the") is frequent. This is a common issue in scientific writing but needs to be addressed for smooth comprehension.

·  Colloquial Language(page 8) Some phrases are too casual for academic writing. Terms like "a lot" or "more easy" should be replaced with formal equivalents (e.g., "considerable" or "easier").

·  Punctuation Errors(page 6) There are a few punctuation issues, particularly in the use of commas and semicolons. Proper punctuation would enhance sentence structure and clarity.

·  Sentence Fragmentation(page 9) Some sentences are overly long and fragmented, making them difficult to follow. Breaking these up into shorter, more concise sentences would improve readability.

·  Transition Phrases(page 10) Transitions between sections and ideas are sometimes abrupt. Adding appropriate transition phrases would improve the flow of the manuscript.

Author Response

3. Point-by-point response to Comments and Suggestions for Authors

  1. Introduction:

Comments 1: The authors should have cited the study by Alnajar et al. (Innovations 2021, DOI: 10.1177/1556984520971775) in both the introduction and discussion, as it directly demonstrates that existing risk scores (e.g., STS-PROM, EuroSCORE II, TAVR-RS) overestimate mortality by 2-to-3-fold for minimally invasive AVR. This aligns with the current study’s focus on refining preoperative stratification for SAVR and would highlight similar findings in risk score inaccuracies, and in the discussion, it would reinforce the argument for recalibrating existing models for newer surgical techniques​.

Response 1: Thank you for pointing this out. We agree with this comment. Indeed, the reference cited was added to the introduction, and information was also added to the discussion such as the potential clinical applicability of our score in minimally invasive surgery. You can see this information between lines 53-55 and 363-366.

  1. Statistics/Methods:

Comments 1: The cohort has STS scores below 4% without a detailed rationale for this exclusion high risk patients. Excluding higher-risk populations can skew comparisons with registry-based risk scores, which reflect the entire spectrum of patient risks​. Including an explanation for this exclusion in the manuscript, and potentially analyzing the outcomes of this excluded group separately, would help clarify whether their mortality rates were significantly different or if they could have been included in the broader analysis.

Response 1: Agree with you. However, I would like to clarify that no patient was excluded, the median scores for both STS and EURO SCORE II were generally of low or intermediate risk in this population, reflecting an underestimation of surgical risk. Therefore, to make the answer clearer, no high-risk patients were excluded.

Comments 2: While the statistical methods used (e.g., Kaplan-Meier, Cox regression) are appropriate for the study, more transparency is needed regarding the failure to normalize certain variables. The manuscript should explain why normalization was not successful and if other methods of transformation could have been used but were not considered for a potential impact to the results.

Response 2: A good comment, thank you. Indeed, we tried to normalize the variables through a logarithmic transformation, but they continued to have an abnormal distribution, so nonparametric statistics were performed to establish the difference between groups. We do not consider that another transformation method would have had an impact on the results. This information was added in the methodology section. (lines 217-219)

Comments 3: Although the authors performed multivariate analysis to adjust for potential confounders, diseases such as diabetes and coronary artery disease were not excluded from the cohort. This introduces a risk of confounding, especially considering their prevalence in the study population. Addressing how these comorbidities may have affected mortality predictions would enhance the robustness of the findings.

Response 3: This is an appropriate observation, thank you very much. Yes, indeed, these are comorbidities that could have an influence; these limitations were included in the text. As we can see in the general table, the 3 groups were quite homogeneous in relation to comorbidities, which relatively reduces this confusion factor. In addition, these variables, such as diabetes and DAC, were included in the multivariate analysis to try to further reduce this risk of confusion. In addition, in relation to DAC, patients with severe coronary disease were excluded. To be concise in the answer, yes, the necessary methodology was used to reduce this bias and the limitations were added to the text as you can see in the lines 369-376.

Comments 4:  The authors mention using SPSS statistical software for data analysis, but further clarification on how missing data was handled would be helpful. It is important to address missing data in retrospective analyses. Were there specific variables missing that might have impacted the accuracy of the risk calculations?

Response 4: Agree. Excellent commentary and insight. Yes, one of the major problems with this type of study is the potential missing variables, however, to reduce this potential problem, only patients with at least >70% of the prespecified variables were included, to reduce this confounding factor. This information was added to the text.

3.     Results

Comments 1: The results section should not be introduced within the methods section. In this manuscript, some descriptions of results (e.g., mortality outcomes and patient demographics) appear prematurely in the methods section, which should be focused solely on the study design, statistical analysis, and procedures. Moving these findings to the results section would improve the manuscript’s structure and ensure clarity and proper flow.

Response 1: Agree. Indeed, but I reviewed the manuscript again, and the mortality results mentioned are not in the methods section, all of those are in the RESULTS section.   

Comments 2: The authors mentioned "prediction" in their methods, but the paper does not seem to include any formal internal validation or predictive modeling to support these claims. Proper prediction modeling typically requires data splitting (e.g., into training and testing sets) or cross-validation. Perhaps I have overlooked it, so please either direct me to the section where prediction modeling was used, or revise the terminology to "association" to reflect the analyses conducted in the study if no prediction modeling was performed.

Response 2: Excellent comment and observation, thank you very much. Yes, there was a confusion when writing, the correct term in this type of analysis is "factors associated with mortality in this population" and not prediction, since there was no validation method. Also, when analyzing the ROC curve, the correct term is the performance capacity of the scale and not predictive analysis. These terms were changed in the text. Thank you very much.

Comments 3: BNP levels showed significant differences across stages, but the multivariate analysis did not find BNP to be an independent predictor. A deeper exploration of why BNP’s predictive value diminished in the multivariate analysis would help clarify its role in the proposed classification system.

Response 3: This is an important, yet complex, question. Based on previous studies that identified BNP as a marker of severity even in asymptomatic patients, our current study did not find this biomarker to be associated with higher mortality. A more plausible explanation could be that structural anatomical changes occur earlier than BNP elevation. This is exemplified in group 2, where BNP levels remained relatively low, yet the group exhibited a notable mortality rate of 10%. This suggests that structural alterations may precede detectable increases in BNP levels. However, within the factors that are associated with very high BNP, only 15% and 6% of the population had EF < 50% and PH, respectively, the latter generally associated with high left filling pressures.

Comments 4:  Although the manuscript states that there were no statistical differences in the ROC curves between the scales, adding more detailed confidence interval comparisons might help clarify the magnitude of the differences. 95% CI will not only show the direction and strength of the demonstrated effect but will also provide a range in which the true value lies.

Response 4: Sorry, I don't know if I understand your question, but those results are detailed in the text, exactly between lines 189-200. Thank you.

Comments 5:  “Complications” was one of the keywords of the manuscript but nor mentioned within the manuscript. In the results, I can only see postop Afib. Reporting major complications such as reentry for bleeding, major wound infections, or paravalvular leaks were not extensively discussed. While the current manuscript focuses heavily on mortality, including these complication rates would add depth to the outcomes, especially in comparison to more traditional SAVR and TAVR techniques, where such complications might be more prevalent.

Response 5: This is an excellent question and would contribute a lot. However, unfortunately for the present work we only have the complications that you have already described. These variables are currently being collected from the same patients and from new ones and the inclusion of these variables will undoubtedly be a great contribution.

4.     Discussion

Comments 1: It would strengthen the discussion to acknowledge that, while the current study focuses on SAVR, the broader surgical landscape includes comparisons to less invasive techniques like mini-AVR and TAVR. These comparisons are critical when discussing mortality and outcomes. For instance, in the mini-AVR population (Alnajar et al. Innovations 2021), it was pointed out that risk scores should ideally reflect differences between traditional and minimally invasive techniques. The authors could also explicitly discuss how their findings might apply to patients undergoing TAVR and if their proposed classification could be generalized to this population

Response 1: Thank you very much for your contribution, yes, those comments have already been added to the introduction and discussion as answered in the previous questions.

Comments 2:  The modified Généreux classification demonstrated similar predictive ability as the original scale, but with fewer stages. However, as this study was performed in a single center in Brazil, thus external validation in larger multicenter datasets, particularly considering that different patient populations may present with different epidemiological characteristics other populations is necessary. Please include a discussion about the generalizability of the model to different ethnic and geographic populations for added important value.

Response 2: Thank you very much for your contribution, yes, this is already added in the discussion mentioned about the clinical applicability in the discussion on lines 289-307, in addition these geographic and epidemiological characteristics were already mentioned previously in the discussion on lines 263-266.

Comments 3:  The study is based on a single-center data, which can introduce bias due to the experience and skill level of individual surgeons. Please address mechanisms in your center for ensuring accurate and complete data capture, including oversight mechanisms and data auditing​. It would be useful particularly since this is a retrospective study and single-institution outcomes can sometimes reflect individual practice nuances.

Response 3: It is an excellent overview. The text is already over the character limit, but if the journal allows me, I could add this information. However, for your information, our institute has extensive surgical experience and a recognized scientific research commission. From this, I can tell you what factors could imply and introduce a bias in the study: because Brazil has a public system and low income, the type of patient who spent a long time waiting in line to perform the intervention is very serious, already with unfavorable evolution of the disease, in addition the volume of surgeries is high and it is a medical training hospital, which may imply when evaluating the final surgical result of the patients. In relation to data capture and medical audit, our institute has a specific area for auditing and controlling electronic patient record data, in addition to the fact that each protocol is rigorously evaluated by the scientific and ethical committee, which gives more value to our data.

Minor Comments:

Limitations Section: Although limitations are discussed, it would benefit from a stronger focus on the potential selection bias due to the retrospective nature of the study. Furthermore, while the study adjusts for some variables, not all potentially important biomarkers (e.g., troponin) were included.

The limitations were modified in the text.

Language and Grammar: Minor grammatical issues are present, such as in the introduction and conclusion. A thorough proofreading would enhance the manuscript’s readability.

The language and grammar were improved.

The authors mentioned using a multivariable cox. I just would like to kindly point out that multivariable ≠ multivariate. This an overwhelmingly common mistake that we all should be aware of in our research. For more clarification please read: Hidalgo and Goodman 2013"Multivariate or multivariable regression?" Am J Public Health doi: 10.2105/AJPH.2012.300897. PMC3518362.

Thank you for your explanation, the article was very helpful, but the word "multivariable cox" is not mentioned in the text, on the contrary, only "multivariate" is mentioned.

Round 2

Reviewer 2 Report

Comments and Suggestions for Authors

Overall, the authors have done an excellent job addressing previous feedback. The revisions made to clarify statistical methods, normalize variables, and adjust the language around predictive modeling were well executed. Below are some minor comments and recommendations for further refinement.

  1. For future studies: The work has improved and future studies will likely to cite it and builds upon its results. I recommend citing the article titled "Commentary: Enhancing risk assessment by incorporating more of what we know" (PMID: 32505455, DOI: 10.1016/j.jtcvs.2020.03.132) in the discussion section to enhance the argument on future directions for improving risk stratification models. This article provides valuable insights into integrating additional variables into surgical risk assessment, aligning with your study's findings.

  2. Missing Data: While the response mentions how missing data was handled, I do not see this explicitly stated in the manuscript or reflected in the tables. In the methods section, please clarify the strategy used for handling missing data (e.g., exclusion criteria, imputation). Additionally, in Table 1, indicate how many values for each variable are unknown or missing to ensure transparency.

  3. Terminology - Multivariable vs. Multivariate: The term "multivariable" should be used instead of "multivariate" throughout the manuscript, I mentioned this the other way around in the previous review. Apologies. This study involves regression models with multiple independent variables; thus, multivariable is the correct terminology accordingly (PMC3518362). 

  4. Regarding the confidence interval comment: Figure 2 will be better displayed with confidence limits as a colored shaded area or bars in the figure is preferred, but both could be added as a table in the figure.

Author Response

3. Point-by-point response to Comments and Suggestions for Authors

Overall, the authors have done an excellent job addressing previous feedback. The revisions made to clarify statistical methods, normalize variables, and adjust the language around predictive modeling were well executed. Below are some minor comments and recommendations for further refinement.

We are very honored to know that our responses were satisfactory and above all grateful for the excellent comments.

Comments 1: For future studies: The work has improved and future studies will likely to cite it and builds upon its results. I recommend citing the article titled "Commentary: Enhancing risk assessment by incorporating more of what we know" (PMID: 32505455, DOI: 10.1016/j.jtcvs.2020.03.132) in the discussion section to enhance the argument on future directions for improving risk stratification models. This article provides valuable insights into integrating additional variables into surgical risk assessment, aligning with your study's findings.

Response 1: Thank you. That article is very interesting. We added that information in the final of discussion section.

 It is important to mention that this article provides another example that there are multiple characteristics that make up the preoperative risk of a patient undergoing cardiac surgery, however, for future studies these preoperative stratification models can improve with some adjustments, as commented by Griffin and colleagues, aiming at greater surveillance in the postoperative period.

Comments 2: Missing Data: While the response mentions how missing data was handled, I do not see this explicitly stated in the manuscript or reflected in the tables. In the methods section, please clarify the strategy used for handling missing data (e.g., exclusion criteria, imputation). Additionally, in Table 1, indicate how many values for each variable are unknown or missing to ensure transparency.

Response 2: Thank you, this information was added in the methods section (statistical analysis):

The criteria used to manage missing data were: apply the exclusion criteria described in the population section, and include only those patients who present at least 70% of the prespecified study variables to reduce possible confounding factors. The missing data of each variable was added and the explanation was added below the table.

Comments 3: Terminology - Multivariable vs. Multivariate: The term "multivariable" should be used instead of "multivariate" throughout the manuscript, I mentioned this the other way around in the previous review. Apologies. This study involves regression models with multiple independent variables; thus, multivariable is the correct terminology accordingly (PMC3518362).

Response 3: Thank you. The word was replaced.

Comments 4: Regarding the confidence interval comment: Figure 2 will be better displayed with confidence limits as a colored shaded area or bars in the figure is preferred, but both could be added as a table in the figure.

These alteration was made and added in the figure 2.
